# Low-Resolution Precoding for Multi-Antenna Downlink Channels and OFDM [note 1]

**DOI:** 10.3390/e24040504

**Published:** 2022-04-04

**Authors:** Andrei Stefan Nedelcu, Fabian Steiner, Gerhard Kramer

**Affiliations:** 1Optical and Quantum Laboratory, Huawei Munich Research Center, 80992 Munich, Germany; andrei.nedelcu2@huawei.com; 2Institute for Communications Engineering, Technical University of Munich (TUM), 80333 Munich, Germany; fabian.steiner@tum.de

**Keywords:** massive MIMO, precoding, coarse quantization, coordinate descent, information rates

## Abstract

Downlink precoding is considered for multi-path multi-input single-output channels where the base station uses orthogonal frequency-division multiplexing and low-resolution signaling. A quantized coordinate minimization (QCM) algorithm is proposed and its performance is compared to other precoding algorithms including squared infinity-norm relaxation (SQUID), multi-antenna greedy iterative quantization (MAGIQ), and maximum safety margin precoding. MAGIQ and QCM achieve the highest information rates and QCM has the lowest complexity measured in the number of multiplications. The information rates are computed for pilot-aided channel estimation and a blind detector that performs joint data and channel estimation. Bit error rates for a 5G low-density parity-check code confirm the information-theoretic calculations. Simulations with imperfect channel knowledge at the transmitter show that the performance of QCM and SQUID degrades in a similar fashion as zero-forcing precoding with high resolution quantizers.

## 1. Introduction

Massive multiple-input multiple-output (MIMO) base stations can serve many UE with high spectral efficiency and simplified signal processing  [1,2]. However, their implementation is challenging due to the cost and energy consumption of analog-to-digital and digital-to-analog converters (ADCs/DACs) and linear PA. There are several approaches to lower cost. One approach is hybrid beamforming with analog beamformers in the RF chain of each antenna and where the digital baseband processing is shared among RF chains. Second, constant envelope waveforms permit using non-linear PA. Third, all-digital approaches use low-resolution ADCs/DACs or low-resolution digitally controlled RF chains. The focus of this paper is on the all-digital approach.

### 1.1. Single-Carrier Transmission

We study the multi-antenna downlink and UE with one antenna each, a model referred to as MU-MISO. Most works on low-cost precoding for MU-MISO consider PSK to lower the requirements on the PA. For instance, the early papers [3,4] (see also [5]) use iterative coordinate-wise optimization to choose transmit symbols from a continuous PSK alphabet for flat and frequency-selective (or multipath) fading, respectively. We remark that these papers do not include an optimization parameter (called α below, see (Equation 8)) in their cost function, which plays an important role at high SNR, see [6,7]. This parameter is related to linear MMSE precoding.

Most works consider discrete alphabet signaling. Perhaps the simplest approach, called QLP, applies a linear precoder followed by one low-resolution quantizer per antenna [8,9,10,11,12,13,14,15]. Our focus is on ZF, and we use the acronyms LP-ZF and QLP-ZF, respectively, for unquantized ZF and the QLP version of ZF.

More sophisticated approaches use optimization tools as in [3,4]. For example, the papers [16,17,18] use convex relaxation methods; Refs. [19,20,21,22,23,24,25] apply coordinate-wise optimization; Refs. [26,27,28] develops a symbol-wise Maximum Safety Margin (MSM) precoder; Refs. [29,30,31,32] use a branch-and-bound (BB) algorithm; Ref. [33] uses a majorization-minimization algorithm; Ref. [34] uses integer programming; and [35,36] use neural networks (NNs). These references are collected in Table 1 together with the papers listed below on OFDM. As the table shows, most papers focus on single-carrier and flat fading channels.

### 1.2. Discrete Signaling and OFDM

Our main interest is discrete-alphabet precoding for multipath channels with OFDM as in 5G wireless systems. Precoding for OFDM is challenging because the alphabet constraint is in the time domain after the inverse discrete Fourier transform (IDFT) rather than in the frequency domain. We further focus on using information theory to derive achievable rates. For this purpose, we consider two types of receivers: classic PAT and a blind detector that performs joint data and channel estimation.

Discrete-alphabet precoding for OFDM was treated in Ref. [37], who used QLP and low resolution DACs. A more sophisticated approach appeared in Ref. [38], who applied a squared-infinity norm Douglas-Rachford splitting (SQUID) algorithm to minimize a quadratic cost function in the frequency domain. The performance was illustrated via BER simulations with convolutional codes and QPSK or 16-QAM by using 1–3 bits of phase quantization.

The paper [39] instead proposed an algorithm called multi-antenna greedy iterative quantization (MAGIQ) that builds on [19] and uses coordinate-wise optimization of a quadratic cost function in the time domain. MAGIQ may thus be considered an extended version of [4] for OFDM and discrete alphabets. Simulations showed that MAGIQ outperforms SQUID in terms of complexity and achievable rates. Another coordinate-wise optimization algorithm appeared in [40,41] that builds on the papers [21,22]. The algorithm is called CESLP and it is similar to the refinement of MAGIQ presented here. The main difference is that, as in [38], the optimization in [40,41] uses a cost function in the frequency domain rather than the time domain. We remark that processing in the time domain has advantages that are described in Section 3.1.

The MSM algorithm was extended to OFDM in [42]. MSM works well at low and intermediate rates but MAGIQ outperforms MSM at high rates both in terms of complexity and achievable rates. Finally, the recent paper [43] uses generalized approximate message passing (GAMP) for OFDM.

### 1.3. Contributions and Organization

The contributions of this paper are as follows.

The analysis of MAGIQ in the workshop paper [39] is extended to larger systems and more realistic channel conditions;Replacing the greedy antenna selection rule of MAGIQ with a fixed (round-robin) schedule is shown to cause negligible rate loss. The new algorithm is named QCM;The performance of QLP-ZF, SQUID, MSM, MAGIQ, and QCM are compared in terms of complexity (number of multiplications and iterations) and achievable rates;We develop an auxiliary channel model to compute achievable rates for pilot-aided channel estimation and a blind detector that performs joint channel and data estimation. The models let one compare modulations, precoders, channels, and receivers;Simulations with a 5G NR LDPC code [44] show that the computed rate and power gains accurately predict the gains of standard channel codes;Simulations with imperfect channel knowledge at the base station show that the achievable rates of SQUID and QCM degrade as gracefully as those of LP-ZF.

We remark that our focus is on algorithms that approximate ZF based on channel inversion, i.e., there is no attempt to optimize transmit powers across subcarriers. This approach simplifies OFDM channel estimation at the receivers because the precoder makes all subcarriers have approximately the same channel magnitude and phase. For instance, a rapid and accurate channel estimate is obtained for each OFDM symbol by averaging the channel estimates of the subcarriers, see Section 4.1. Of course, it is interesting to develop algorithms for other precoders and for subcarrier power allocation.

This paper is organized as follows. Section 2 introduces the baseband model and OFDM signaling. Section 3 describes the MAGIQ and QCM precoders. Section 4 develops theory for achievable rates, presents complexity comparisons, and reviews a model for imperfect CSI. Section 5 compares achievable rates and BER with 5G NR LDPC codes. Section 6 concludes the paper.

## 2. System Model

Figure 1 shows a MU-MISO system with *N* transmit antennas and *K*UE that each have a single antenna. The base station has one message per UE and each antenna has a resolution of 1 bit for the amplitude (on-off switch) and *b* bits for the phase per antenna. All other hardware components are ideal: linear, infinite bandwidth, no distortions except for AWGN.

### 2.1. Baseband Channel Model

The discrete-time baseband channel is modeled as a finite impulse response filter between each pair of transmit and receive antennas. Let $x_{n}\left[t\right]$ be the symbol of transmit antenna *n* at time *t* and let $\left[t\right]={\left(x_{1}\left[t\right]\;\cdots \;x_{N}\left[t\right]\right)}^{T}$. Similarly, let $y_{k}\left[t\right]$ be the received symbol of UE *k* at time *t* and let $\left[t\right]={\left(y_{1}\left[t\right]\;\cdots \;y_{K}\left[t\right]\right)}^{T}$. The channel model is
(1)$$\begin{array}{c} y[t]=\sum_{\tau =0}^{L-1}H[\tau ]x[t-\tau ]+z[t] \end{array}$$
where the noise $\left[t\right]={\left(z_{1}\left[t\right]\;\cdots \;z_{K}\left[t\right]\right)}^{T}$ has circularly-symmetric, complex, Gaussian entries that are independent and have variance $\sigma^{2}$, i.e., we have $z∼CN(0,\sigma^{2}I)$. The [τ], τ=0,⋯,L−1, are K×N matrices representing the channel impulse response, i.e., we have
(2)$$\left[\tau \right]=\left(\begin{array}{cccc} h_{11}\left[\tau \right] & h_{12}\left[\tau \right] & \ldots \; & h_{1N}\left[\tau \right]\\ h_{21}\left[\tau \right] & h_{22}\left[\tau \right] & \ldots \; & h_{2N}\left[\tau \right]\\ \vdots & \vdots & \ddots \; & \vdots \\ h_{K1}\left[\tau \right] & h_{K2}\left[\tau \right] & \ldots \; & h_{KN}\left[\tau \right] \end{array}\right)$$
where $h_{kn}\left[.\right]$ is the channel impulse response from the *n*-th antenna at the base station to the *k*-th UE. For instance, a Rayleigh fading multi-path channel with a uniform PDP has $h_{kn}\left[\tau \right]∼CN\left(0,1/L\right)$ and these taps are independent and identically
distributed (iid) for all k,n,τ.

The vector [t] is constrained to have entries taken from a discrete and finite alphabet
(3)$$𝒳=\left\{0\right\}\cup \left\{\sqrt{\frac{P}{N}}\;e^{2\pi q/2^{b}};q=0,1,2,\cdots ,2^{b}-1\right\}.$$

The transmit energy clearly satisfies $∥x\left[t\right]{∥}^{2}\leq P$ and we define $\mathrm{SNR}=P/\sigma^{2}$. The inequality is due to the 0 symbol that permits antenna selection. Antenna selection was also used in [45] to enforce sparsity. Our intent is rather to allow antennas not to be used if they do not improve performance.

### 2.2. OFDM Signaling

Figure 1 shows how OFDM can be combined with the precoder. Let $T=T_{F}+T_{c}$ be the OFDM blocklength with $T_{F}$ symbols for the DFT and $T_{c}$ symbols for the cyclic prefix. We assume that $T_{F}\geq L$ and $T_{c}\geq L-1$. For simplicity, all $T_{F}$ subcarriers carry data and we do not include the cyclic prefix overhead in our rate calculations below, i.e., the rates in bpcu are computed by normalizing by $T_{F}$.

Consider the frequency-domain modulation alphabet $\hat{U}$ that has a finite number of elements, e.g., QPSK has $\hat{U}=\{\hat{u}:\hat{u}=\left(\pm 1\;\pm j\right)/\sqrt{2}\}$. Messages are mapped to the frequency-domain vectors $\hat{u}\left[m\right]=({\hat{u}}_{1}\left[m\right],\cdots ,{\hat{u}}_{K}\left[m\right]{)}^{T}$ for subcarriers $m=0,\cdots ,T_{F}-1$ that are converted to time-domain vectors [t] by IDFTs
(4)$$u_{k}\left[t\right]=\frac{1}{T_{F}}\sum_{m=0}^{T_{F}-1}{\hat{u}}_{k}\left[m\right]e^{2\pi mt/T_{F}}$$
for times $t=0,\cdots ,T_{F}-1$ and UE k=1,⋯,K. For the simulations below, we generated the ${\hat{u}}_{k}\left[m\right]$ uniformly from finite constellations such as 16-QAM or 64-QAM. We assume that $E[{\hat{u}}_{k}\left[m\right]]=0$ for all *k* and *m*. Each UE *k* uses a DFT to convert its time-domain symbols $y_{k}\left[t\right]$ to the frequency-domain symbols
(5)$${\hat{y}}_{k}\left[m\right]=\sum_{t=0}^{T_{F}-1}y_{k}\left[t\right]e^{-2\pi mt/T_{F}}.$$

### 2.3. Linear MMSE Precoding

To describe the linear MMSE precoder, consider the channel from base station antenna *n* to UE *k*: (6)$$\begin{array}{c} h_{kn}=\left(h_{kn}\left[0\right],\cdots ,h_{kn}\left[L-1\right],\underset{\left(T_{F}-L\right)\;\mathrm{zeros}}{\underset{︸}{0,\cdots ,0}}\right) \end{array}$$
and denote its DFT as ${\hat{h}}_{kn}=({\hat{h}}_{kn}\left[0\right],\cdots ,{\hat{h}}_{kn}\left[T_{F}-1\right]{)}^{T}$. The channel of subcarrier *m* is the K×N matrix $\hat{H}\left[m\right]$ with entries ${\hat{h}}_{kn}\left[m\right]$ for k=1,⋯,K, n=1,⋯,N. The linear MMSE precoder (or Wiener filter) for subcarrier *m* is
(7)$$\begin{array}{c} P\left[m\right]\hat{H}{\left[m\right]}^{†}{\left(P\left[m\right]\hat{H}\left[m\right]\hat{H}{\left[m\right]}^{†}+\sigma^{2}I\right)}^{-1} \end{array}$$
where $P\left[m\right]=E[|{\hat{u}}_{k}\left[m\right]{|}^{2}]$ is the same for all *k*, $\hat{H}{\left[m\right]}^{†}$ is the Hermitian of $\hat{H}\left[m\right]$, and is the K×K identity matrix. The precoder multiplies $\hat{u}\left[m\right]$ by (Equation 7) for all subcarriers *m*, and performs *N* IDFTs to compute the resulting $x\left[0\right],\cdots ,x\left[T_{F}-1\right]$. We remark that ZF precoding is the same as (Equation 7) but with $\sigma^{2}=0$, where $\hat{H}\left[m\right]\hat{H}{\left[m\right]}^{†}$ is usually invertible if *N* is much larger than *K*.

## 3. Quantized Precoding

We wish to ensure compatibility with respect to LP-ZF. In other words, each receiver *k* should ideally see signals $u_{k}\left[t\right]$, t=0,⋯,T−1, that were generated from the frequency-domain signals ${\hat{u}}_{k}\left[m\right]$, $m=0,\cdots ,T_{F}-1$. Let $\left[t\right]={\left(u_{1}\left[t\right]\;\cdots \;u_{K}\left[t\right]\right)}^{T}$ and define the time-domain mean square error (MSE) cost function
(8)$$\begin{array}{cc} G(x[0],\cdots ,x[T-1],\alpha )\; & =\sum_{t=0}^{T-1}E_{\left[t\right]}\left[{\left\|u[t]-\alpha y[t]\right\|}^{2}\right]\\ \; & =\sum_{t=0}^{T-1}{\left\|u[t]-\alpha \sum_{\tau =0}^{L-1}H[\tau ]x[t-\tau ]\right\|}^{2}+\alpha^{2}TK\sigma^{2} \end{array}$$
where $E_{\left[t\right]}\left[\cdot \right]$ denotes the expectation with respect to the noise z[t]. The optimization problem is as follows:(9)$$\begin{array}{cccc} \; & \min_{x[0],\cdots ,x[T-1],\;\alpha }\; & \; & G(x[0],\cdots ,x[T-1],\alpha )\\ \; & s.t. & \; & x\left[t\right]\in {𝒳}^{N},\;t=0,\cdots ,T-1\\ \; & \; & & \alpha >0. \end{array}$$

The parameter α in (Equation 8) and (Equation 9) can easily be optimized for fixed x[0],⋯,x[T−1] and the result is (see [18] Equation (26))
(10)α=∑t=0T−1Reu[t]H∑τ=0L−1H[τ]x[t−τ]∑t=0T∥∑τ=0L−1H[τ]x[t−τ]∥2+TKσ2.

For the MAGIQ and QCM algorithms below, we use alternating minimization to find the x[0],⋯,x[T−1] and α. For the linear MMSE precoder, we label the α in (Equation 10) as $\alpha_{\mathrm{WF}}$.

Observe that we use the same α for all *K*UE because all UE experience the same shadowing, i.e., all *K*UE see the same average power. For UE-dependent shadowing, a more general approach would be to replace α with a diagonal matrix with *K* parameters $\alpha_{k}$, k=1,⋯,K, and then modify (Equation 8) appropriately.

### 3.1. MAGIQ and QCM

For multipath channels, the vector x[t] influences the channel output at times *t*,
t+1, ⋯,t+L−1. A joint optimization over strings of length *T* seems difficult because of this influence and because of the finite alphabet constraint for the $x_{n}\left[t\right]$. Instead, MAGIQ splits the optimization into sub-problems with reduced complexity by applying coordinate-wise minimization across the antennas and iterating over the OFDM symbol.

For this purpose, consider the precoding problem for time $t^{\prime }$ starting at $t^{\prime }=0$ and ending at $t^{\prime }=T-1$. Observe that $x[t^{\prime }]$ influences at most *L* summands in (Equation 8), namely the summands for $t={\left(t^{\prime }\right)}_{T},\cdots ,{\left(t^{\prime }+L-1\right)}_{T}$ where ${\left(t\right)}_{T}=\min\left(t,T-1\right)$. To compute the new cost after updating the symbol $x_{n}\left[t^{\prime }\right]$, one may thus compute sums of the form
(11)$$\begin{array}{c} \sum_{t={\left(t^{\prime }\right)}_{T},\cdots ,{\left(t^{\prime }+L-1\right)}_{T}}{\left\|u\left[t\right]-\alpha \sum_{\tau =0}^{L-1}H[\tau ]x\left[t-\tau \right]\right\|}^{2} \end{array}$$
for $t^{\prime }=0,\cdots ,T-1$. In both cases, one computes a first and second sum having the old and new $x_{n}\left[t^{\prime }\right]$, respectively. One then takes the difference and adds the result to (Equation 8) to obtain the updated cost.

We remark that the time-domain cost function (Equation 8) is closely related to the frequency-domain cost functions in [38,40,41]. However, the time-domain approach is more versatile as it can include acyclic phenomena such as interference from previous OFDM blocks. The time-domain approach is also slightly simpler because updating the symbol $x_{n}\left[t^{\prime }\right]$ in (Equation 8) or (Equation 11) requires taking the norm of at most *L* vectors of dimension *K* for each test symbol in X while the frequency-domain approach in ([40] Equation (Equation 17)) takes the norm of $T_{F}$ vectors of dimension *K* for each test symbol. Recall that $T_{F}\geq L$, and usually $T_{F}\geq 10L$ to avoid losing too much efficiency with the cyclic prefix that has length $T_{c}\geq L-1$.

The MAGIQ algorithm is summarized in Algorithm 1. MAGIQ steps through time in a cyclic fashion for fixed α. At each time *t*, it initializes the antenna set ={1,⋯,N} and performs a greedy search for the antenna *n* and symbol $x_{n}\left[t\right]$ that minimize (Equation 8) (one may equivalently consider sums of *L* norms as in (Equation 11)). The resulting antenna is removed from and a new greedy search is performed to find the antenna in the new and the symbol that minimizes (Equation 8) while the previous symbol assignments are held fixed. This step is repeated until is empty. MAGIQ then moves to the next time and repeats the procedure. To determine α, MAGIQ applies alternating minimization with respect to α and the precoder output x{[t]:t=0,⋯,T−1}. For fixed [.] the minimization can be solved in closed form, see (Equation 10) and line 22 of Algorithm 1.

Simulations show that MAGIQ exhibits good performance and converges quickly [39]. However, the greedy selection considerably increases the computational complexity. We thus replace the minimization over in line 9 of Algorithm 1) with a round-robin schedule or a random permutation. We found that both approaches perform equally well. The new QCM algorithm performs as well as MAGIQ but with a simpler search and a small increase in the number of iterations.

Finally, one might expect that α is close to the $\alpha_{\mathrm{WF}}$ of the transmit Wiener filter [6,7] since our cost function accounts for the noise power. However, Figure 2 shows that this is true only at low SNR. The figure plots the average α of the QCM algorithm, called $\alpha_{\mathrm{QCM}}$, against the computed $\alpha_{\mathrm{WF}}$ for simulations with System A in Section 5. Note that $\alpha_{\mathrm{QCM}}$ is generally larger than $\alpha_{\mathrm{WF}}$.
**Algorithm 1:**MAGIQ and QCM precoding.1:**procedure**Precode(Algo, H[.], u[.])2:    $x^{\left(0\right)}\left[.\right]={x[.]}_{init}$3:    $\alpha^{\left(0\right)}=\alpha_{init}$4:    **for** i=1:I **do** // iterate over OFDM block5:        **for** t=0:T−1 **do**6:           ={1,…,N}7:           **while** S≠⌀ **do**8:               **if** Algo = MAGIQ **then**9:                   $\left(x_{n^{☆}}^{☆},n^{☆}\right){=}_{{\mathrm{argmin}}_{{\tilde{x}}_{n}\in 𝒳,n\in S}}$10:                   $\;\;G\left(x^{\left(i\right)}\left[0\right],\cdots ,x^{\left(i\right)}\left[t-1\right],\tilde{x},\right.$11:                   $\left.\;x^{\left(i-1\right)}\left[t+1\right],\cdots ,x^{\left(i-1\right)}\left[T-1\right],\alpha^{\left(i-1\right)}\right)$12:               **else** // Algo = QCM13:                   $n^{☆}=\min\;S$ // round-robin schedule14:                   $x_{n^{☆}}^{☆}={\mathrm{argmin}}_{{\tilde{x}}_{n^{☆}}\in 𝒳}$15:                   $\;\;G\left(x^{\left(i\right)}\left[0\right],\cdots ,x^{\left(i\right)}\left[t-1\right],\tilde{x},\right.$16:                   $\left.\;x^{\left(i-1\right)}\left[t+1\right],\cdots ,x^{\left(i-1\right)}\left[T-1\right],\alpha^{\left(i-1\right)}\right)$17:               **end if**18:               $x_{n^{☆}}^{\left(i\right)}\left[t\right]=x_{n^{☆}}^{☆}$ // update antenna $n^{☆}$ at time *t*19:               $S\leftarrow S\setminus \{n^{☆}\}$20:           **end while**21:        **end for**22:        α(i)=∑t=0T−1Reu[t]H∑τ=0L−1H[τ]x(i)[t−τ]∑t=0T∑τ=0L−1H[τ]x(i)[t−τ]2+TKσ223:    **end for**24:    **return** $x[.]=x^{\left(I\right)}\left[.\right],\alpha =\alpha^{\left(I\right)}$25:**end procedure**

## 4. Performance Metrics

### 4.1. Achievable Rates

We use GMI to compute achievable rates [46,47], (Ex. 5.22) which is a standard tool to compare coded systems. Consider a generic input distribution P(x) and a generic channel density p(y|x) where $={\left(x_{1},\cdots ,x_{S}\right)}^{T}$ and $={\left(y_{1},\cdots ,y_{S}\right)}^{T}$ each have *S* symbols. A lower bound to the mutual information
(12)$$I(X;Y)=\sum_{x,y}P\left(x\right)p\left(y|x\right)\log_{2}\left(\frac{p(y|x)}{\sum_{a}P\left(a\right)\;p\left(y|a\right)}\right)$$
is the GMI
(13)$$\begin{array}{c} I_{q,s}\left(X;Y\right)=\sum_{x,y}P\left(x\right)p\left(y|x\right)\log_{2}\left(\frac{q{\left(y|x\right)}^{s}}{\sum_{a}P\left(a\right)\;q{\left(y|x\right)}^{s}}\right) \end{array}$$
where q(y|x) is any auxiliary density and s≥0. In other words, the choices q(y|x)=p(y|x) for all, and s=1 maximize the GMI. However, the idea is that p(y|x) may be unknown or difficult to compute and so one chooses a simple q(y|x). The reason why p(y|x) is difficult to compute here is because we will measure the GMI across the end-to-end channels from the ${\hat{u}}_{k}\left[m\right]$ to the ${\hat{y}}_{k}\left[m\right]$ and the quantized precoding introduces non-linearities in these channels. The final step in evaluating the GMI is maximizing over s≥0. Alternatively, one might wish to simply focus on s=1, e.g., see [48].

We study the GMI of two non-coherent systems: classic PAT and a blind detector that performs joint data and channel estimation. For both systems, we apply memoryless signaling with the product distribution
(14)$$\begin{array}{c} P\left(x\right)=\prod_{i=1}^{S_{p}}1\left(x_{i}=x_{p,i}\right)\cdot \prod_{i=S_{p}+1}^{S}P\left(x_{i}\right) \end{array}$$
where the $x_{p,i}$ are pilot symbols, 1(a=b) is the indicator function that takes on the value 1 if its argument is true and 0 otherwise, and P(x) is a uniform distribution. Joint data and channel estimation has $S_{p}=0$ so that we have only the second product in (Equation 14). At the receiver we use the auxiliary channel
(15)$$\begin{array}{c} q\left(y|x\right)=\prod_{i=1}^{S}q_{,}\left(y_{i}\;|\;x_{i}\right) \end{array}$$
where the symbol channel $q_{x,y}\left(.\right)$ is a function of ***x*** and ***y***. Observe that $q_{x,y}\left(.\right)$ is invariant for *S* symbols and the channel can be considered to have memory since every symbol $x_{\ell }$ or $y_{\ell }$, ℓ=1,⋯,S, influences the channel for all “times” i=1,⋯,S. The GMI rate (Equation 13) simplifies to
(16)$$\begin{array}{c} \sum_{x,y}P\left(x\right)p\left(y|x\right)\sum_{i=S_{p}+1}^{S}\log_{2}\left(\frac{q_{x,y}{\left(y_{i}\;|\;x_{i}\right)}^{s}}{\sum_{a}P\left(a\right)\;q_{x,y}{\left(y_{i}\;|\;a\right)}^{s}}\right). \end{array}$$

One may approximate (Equation 16) by applying the law of large numbers for stationary signals and channels. The idea is to independently generate the *B* pairs of vectors
$$\begin{array}{cc} \; & x^{\left(b\right)}={\left(x_{1}^{\left(b\right)},\cdots ,x_{S}^{\left(b\right)}\right)}^{T}\\ \; & y^{\left(b\right)}={\left(y_{1}^{\left(b\right)},\cdots ,y_{S}^{\left(b\right)}\right)}^{T} \end{array}$$
for b=1,⋯,B, and then the following average rate will approach $I_{q,s}\left(X;Y\right)/S$bpcu as *B* grows: (17)$$\begin{array}{c} R_{a}=\frac{1}{B}\sum_{b=1}^{B}R_{a}^{\left(b\right)} \end{array}$$
where
(18)$$\begin{array}{c} R_{a}^{\left(b\right)}=\frac{1}{S}\sum_{i=S_{p}+1}^{S}\log_{2}\left(\frac{q_{x^{\left(b\right)},y^{\left(b\right)}}{\left(y_{i}^{\left(b\right)}\;|\;x_{i}^{\left(b\right)}\right)}^{s}}{\sum_{a}P\left(a\right)\;q_{x^{\left(b\right)},y^{\left(b\right)}}{\left(y_{i}^{\left(b\right)}\;|\;a\right)}^{s}}\right). \end{array}$$

We choose the Gaussian auxiliary density
(19)$$q_{x,y}\left(y|x\right)=\frac{1}{\pi \sigma_{q}^{2}}\exp\left(-\frac{{\left|y-h\cdot x\right|}^{2}}{\sigma_{q}^{2}}\right)$$
where for PAT the receiver computes joint ML estimates with sums of $S_{p}$ terms:(20)$$\begin{array}{c} \begin{array}{cc} \; & h=\frac{\sum_{i=1}^{S_{p}}y_{i}\cdot x_{i}^{*}}{\sum_{i=1}^{S_{p}}\left|x_{i}^{2}\right|}\\ \; & \sigma_{q}^{2}=\frac{1}{S_{p}}\sum_{i=1}^{S_{p}}{\left|y_{i}-h\cdot x_{i}\right|}^{2}. \end{array} \end{array}$$

For the blind detector we replace $S_{p}$ with *S* in (Equation 20). Note that for the Gaussian channel (Equation 19) the parameter *s* multiplies $1/\sigma_{q}^{2}$ in (Equation 16) or (Equation 18), and optimizing *s* turns out to be the same as choosing the best parameter $\sigma_{q}^{2}$ when s=1.

Summarizing, we use the following steps to evaluate achievable rates. Suppose the coherence time is $S/T_{F}$ OFDM symbols where *S* is a multiple of $T_{F}$. We index the channel symbols by the pairs (ℓ,m) where *ℓ* is the OFDM symbol and *m* is the subcarrier, $1\leq \ell \leq S/T_{F}$, 0≤m≤T−1. We collect the pilot index pairs in the set $S_{p}$ that has cardinality $S_{p}$, and we write the channel inputs and outputs of UE *k* for OFDM symbol *ℓ* and subcarrier *m* as ${\hat{u}}_{k}\left[\ell ,m\right]$ and ${\hat{y}}_{k}\left[\ell ,m\right]$, respectively.

(1)Repeat the following steps (2)–(4) *B* times; index the steps by b=1,⋯,B;(2)Use Monte Carlo simulation to generate the symbols $H{\hat{u}}_{k}\left[\ell ,m\right]$ and ${\hat{y}}_{k}\left[\ell ,m\right]$ for k=1,⋯,K, $\ell =1,\cdots ,S/T_{F}$, and m=0,⋯,T−1;(3)Each UE estimates its own channel $h_{k}$ and $\sigma_{q,k}^{2}$, i.e., the channel estimate (Equation 20) of UE *k* is
(21)$$\begin{array}{c} \begin{array}{cc} \; & h_{k}=\frac{\sum_{\left(\ell ,m\right)\in S_{p}}{\hat{y}}_{k}\left[\ell ,m\right]\cdot {\hat{u}}_{k}{\left[\ell ,m\right]}^{*}}{\sum_{\left(\ell ,m\right)\in S_{p}}{\left|{\hat{u}}_{k}\left[\ell ,m\right]\right|}^{2}}\\ \; & \sigma_{q,k}^{2}=\frac{1}{S_{p}}\sum_{\left(\ell ,m\right)\in S_{p}}{\left|{\hat{y}}_{k}\left[\ell ,m\right]-h_{k}\cdot {\hat{u}}_{k}\left[\ell ,m\right]\right|}^{2}. \end{array} \end{array}$$For the blind detector, in (Equation 21) we replace $S_{p}$ with the set of all index pairs (ℓ,m), and we replace $S_{p}$ with *S*;(4)Compute $R^{\left(b\right)}$ in (Equation 18) for each UE *k* by averaging, i.e., the rate for UE *k* is
(22)$$\begin{array}{c} R_{a,k}^{\left(b\right)}=\frac{1}{S}\sum_{\left(\ell ,m\right)\notin S_{p}}\log_{2}\left(\frac{q_{{\hat{u}}_{k},{\hat{y}}_{k}}{\left({\hat{y}}_{k}\left[\ell ,m\right]\;|\;{\hat{u}}_{k}\left[\ell ,m\right]\right)}^{s}}{\sum_{a}P\left(a\right)\;q_{{\hat{u}}_{k},{\hat{y}}_{k}}{\left({\hat{y}}_{k}\left[\ell ,m\right]\;|\;a\right)}^{s}}\right) \end{array}$$
where ${\hat{u}}_{k}$ and ${\hat{y}}_{k}$ are vectors collecting the ${\hat{u}}_{k}\left[\ell ,m\right]$ and ${\hat{y}}_{k}\left[\ell ,m\right]$, respectively, for all pairs (ℓ,m). For the blind detector we set $S_{p}=\emptyset$ in (Equation 22);(5)Compute *R*_a_ in (Equation 17) for each UE, i.e., the average rate of UE *k* is $R_{a,k}=\frac{1}{B}\sum_{b=1}^{B}R_{a,k}^{\left(b\right)}$;(6)Compute the average UE rate ${\bar{R}}_{}=\frac{1}{K}\sum_{k=1}^{K}R_{a,k}$.

Our simulations showed that optimizing over s≥0 gives s≈1 if the channel parameters are chosen using (Equation 21).

### 4.2. Discussion

We make a few remarks on the lower bound. First, the receivers do not need to know α. Second, the rate *R*_a_ in (Equation 17) is achievable if one assumes stationarity and coding and decoding over many OFDM blocks. Third, as *S* grows, the channel estimate of the blind detector becomes more accurate and the performance approaches that of a coherent receiver. Related theory for PAT and large *S* is developed in [49]. However, the PAT rate is generally smaller than for a blind detector because the PAT channel estimate is less accurate and because PAT does not use all symbols for data.

Next, let $\left|\hat{U}\right|$ be the number of elements in the modulation set ${\left|\hat{U}\right|}^{S}$. The blind detector must generate ${\left|\hat{U}\right|}^{S}$ likelihoods, which is prohibitively large unless $\left|\hat{U}\right|$ and *S* are small. Moreover, the receiver must perform joint data and channel estimation. Blind detection algorithms can, e.g., be based on high-order statistics and iterative channel estimation and decoding. For polar codes and low-order constellations, one may use the blind algorithms proposed in [50]. We found that the PAT rates are very close (within 0.1 bpcu) of the pilot-free rates multiplied by the rate loss factor $1-S_{p}/S$ for pilot fractions as small as $S_{p}/S=10\%$.

Depending on the system under consideration, we choose one of $T_{F}=32,256,396$, one of T=35,270,277,286,410, one of S=256,1584, and B=200. For most simulations we have $T_{F}=S=256$ and estimate the channel based on individual OFDM symbols, see Section 1.3. For example, for T=270 and a symbol time of 30 ns (symbol rate 33.3 MHz) the coherence time needs to be at least (30ns)·T=8.1 μs. Of course, the transmitter needs to know the channel also, e.g., via time-division duplex, which requires the coherence time to be substantially larger. The main point is that channel estimation at the receiver is not a bottleneck when using ZF based on channel inversion. Finally, for the coded simulations we chose $T_{F}=396$ and $S=4T_{F}=1548$ because the LDPC code occupies four OFDM symbols.

### 4.3. Algorithmic Complexity

This section studies the algorithmic complexity in terms of the number of multiplications and iterations. The complexity of SQUID is thoroughly discussed in [38] and Table 3 shows the order estimates take from [38] (Table I). Note the large number of iterations.

The complexity of MSM depends on the choice of optimization algorithm and [42] considers a simplex algorithm. Unfortunately, the simplex algorithm requires a large number of iterations to converge because this number is proportional to the number of variables and linear inequalities that grow with the system size (N,K,T). An interior point algorithm converges more quickly but has a much higher complexity per iteration.

For MAGIQ and QCM, Equation (Equation 8) shows that updating x[.] requires updating *L* of the *T* terms that each require a norm calculation. The resulting terms ${∥u\left[t\right]∥}^{2}$ do not affect the maximization; terms such as ${∥\alpha Hx∥}_{2}^{2}$ can be pre-computed and stored with a complexity of NKL|𝒳|, and then reused as they do not change during the iterations. On the other hand, products of the form $\alpha u^{H}Hx$ must be computed for each of the *L* terms for each antenna update and at each time instance, resulting in a complexity of O(NKLT). The initialization requires KNT multiplications and one must transform the solutions to the time domain. We neglect the cost of updating α because the terms needed to compute it are available as a byproduct of the iterative process over the time instances.

### 4.4. Sensitivity to Channel Uncertainty at the Transmitter

In practice, the CSI is imperfect due to noise, quantization, calibration errors, etc. We do not attempt to model these effects exactly. Instead, we adopt a standard approach based on MMSE estimation and provide the precoder with channel matrices $\tilde{H}\left[\tau \right]$ that satisfy
(23)$$\begin{array}{c} H\left[\tau \right]=\sqrt{1-\varepsilon^{2}}\tilde{H}\left[\tau \right]+\varepsilon Z\left[\tau \right] \end{array}$$
where 0≤ε≤1 and Z[τ] is a K×N matrix of independent, variance $\sigma_{h}^{2}=1/L$, complex, circularly-symmetric Gaussian entries. Note that ε=0 corresponds to perfect CSI and ε=1 corresponds to no CSI. The precoder treats $\tilde{H}\left[\tau \right]$ as the true channel realization for τ=0,⋯,L−1.

## 5. Numerical Results

We evaluate the GMI of four systems. The main parameters are listed in Table 2 and we provide a few more details here.

System A: the DFT has length $T_{F}=256$ and the channel has either L=1 or L=15 taps of Rayleigh fading with a uniform PDP. The minimum cyclic prefix length for the latter case is $T_{c}=14$ so the minimum OFDM blocklength is T=270;System B: MSM is applied to PSK. However, the MSM complexity limited the simulations to smaller parameters than for System A. The channel now has L=4 taps of Rayleigh fading with a uniform PDP. The T=35 OFDM symbols include a DFT of length $T_{F}=32$ and a minimum cyclic prefix length of $T_{c}=3$;System C: System C is actually two systems because we compare the performance under Rayleigh fading to the performance with the Winner2 model [51] whose number *L* of channel taps varies randomly. For the Winner2 channel, the choice $T_{c}=30$ suffices to ensure that $T_{c}\geq L-1$. The Rayleigh fading model has L=22 taps with a uniform PDP, where *L* was chosen as the maximum Winner2 channel length that has almost all the channel energy;System D: similar to System A but for a 5G NR LDPC code with code rate 8/9 and 64-QAM for an overall rate of 5.33 bpcu. The LDPC code uses the BG1 base graph of the 3GPP Specification 38.212 Release 15, including puncturing and shortening as specified in the standard. The code length is 9504 bits or 1584 symbols of 64-QAM; this corresponds to 4 frames of $T_{F}=396$ symbols.The codewords were transmitted using at least T=410 symbols that include a DFT of length $T_{F}=396$ and a minimum cyclic prefix length of $T_{c}=14$.

The average GMI for Systems A–C were computed using S=256, B=200, and a blind detector. The coded results of System D instead have S=1584 symbols to fit the block structure determined by the LDPC encoder. For System D we considered both PAT and a blind detector. For all cases, the GMI was computed by averaging over the sub-carriers, i.e., channel coding is assumed to be applied over multiple sub-carriers and OFDM symbols. The MAGIQ and QCM algorithms were both initialized with a time-domain quantized solution of the transmit matched filter (MF).

Figure 3 and Figure 4 show the average GMI for System A with b=2 and b=3, respectively. In Figure 3, MAGIQ performs four iterations for each OFDM symbol while QCM performs six iterations. Observe that MAGIQ and QCM are best at all SNR and they are especially good in the interesting regime of high SNR and rates. The gap to the rates over flat fading channels (L=1) is small. SQUID with 64-QAM requires 100–300 iterations for SNR>15 dB and a modified algorithm with damped updates, otherwise SQUID diverges. In addition, we show the broadcast channel capacity with uniform power allocation and Gaussian signaling as an upper bound for the considered scenario [52,53]. Figure 4 shows that QCM with three iterations operates within ≈0.2–0.4 dB of MAGIQ with five iterations when b=3, which shows that QCM performs almost as well as MAGIQ.

Figure 5 compares achievable rates of QCM, SQUID, and MSM for a smaller system studied in [42]. We use PSK because the MSM algorithm was designed for PSK. The figure shows that MSM outperforms SQUID and QCM at low to intermediate SNR and rates, but QCM is best at high SNR and rates. This suggests that modifying the cost function (Equation 8) to include a safety margin will increase the QCM rate at low to intermediate SNR, and similarly modifying the MSM optimization to more closely resemble QCM will increase the MSM rate at high SNR. We tried to simulate MSM for System A but the algorithm ran into memory limitations (we used 2 AMD EPYC 7282 16-Core processors, 125 GB of system memory, and Matlab with both dual-simplex and interior-point solvers).

Consider next the Winner2 non-line-of-sight (NLOS) C2 urban model [51], which is more realistic than Rayleigh fading. The model parameters are as follows.

Base station at the origin (x,y)=(0,0);100 drops of 8 UE placed on a disk of radius 150 m centered at (x,y)=(0,200m); the locations of the UE are iid with a uniform distribution on the disc;8 × 10 uniform rectangular antenna array at the base station with half-wavelength dipoles at λ/2 spacing;5 MHz bandwidth at center frequency 2.53 GHz;No Doppler shift, shadowing and pathloss.

Figure 6 shows the average GMI for LP-ZF and MAGIQ. At high SNR, there is a slight decrease in the slope of the MAGIQ GMI as compared to LP-ZF. This suggests that one might need a larger *N* or *b*. The performance for the Rayleigh fading model is better than for the Winner2 model but otherwise behaves similarly.

Figure 7 shows BER for the LDPC code with 64-QAM. Each codeword is interleaved over 4 OFDM symbols, all 396 subcarriers, and the 6 bits of each modulation symbol by using bit-interleaved coded modulation (BICM). The interleaver was chosen randomly with a uniform distribution over all permutations of length 9504. The solid curves are based on estimating the channel with the transmitted symbols, i.e., these curves are for a genie-aided channel estimator and give lower bounds on the performance of a blind detector. The dotted curves show the performance of PAT when the fraction of pilots is $S_{p}/S=10\%$. The pilots were placed uniformly at random over the four OFDM symbols and 396 subcarriers. A good blind detector algorithm that performs joint channel and data estimation should have BER between the solid and dotted curves.

The dashed curves in Figure 7 show the SNR required for the different algorithms based on Figure 3. In particular, the rate 5.33 bpcu requires SNR of 9 dB, 12.9 dB, and 15.2 dB for LP-ZF, QCM, and SQUID, respectively. SQUID is run with 300 iterations and QCM is run with 6 iterations. Each UE computes its log-likelihoods based on the parameters (Equation 20) of the auxiliary channel. The GMI predicts the coded behavior of the system within approximately 1 dB of the code waterfall region, except for SQUID, where the gap is about 2 dB. The gap seems to be caused mainly by the finite-blocklength of the LDPC code, since the smaller gap of approximately 1 dB is also observed for additive white Gaussian noise (AWGN) channels. The sizes of the gaps are different, and the reason may be that the slopes of the GMI at rate 5.33 bpcu are different, see Figure 3. Observe that LP-ZF exhibits the steepest slope and SQUID the flattest at $R_{a}=5.33$ bpcu; this suggests that SQUID’s SNR performance is more sensitive to the blocklength.

Figure 8 is for System A and shows how the GMI decreases as the CSI becomes noisier. The behavior of all systems is qualitatively similar. However, the figure shows that the QCM rate is more sensitive to the parameter ϵ than the SQUID rate when ϵ is small.

## 6. Conclusions

We studied downlink precoding for MU-MISO channels where the base station uses OFDM and low-resolution DACs. A QCM algorithm was introduced that is based on the MAGIQ algorithm in [39] (see also [19]) and which performs a coordinate-wise optimization in the time-domain. The performance was analyzed by computing the GMI for two auxiliary channel models: one model for pilot-aided channel estimation and a second model for a blind detector that performs joint channel and data estimation. Simulations for several downlink channels, including a Winner2 NLOS urban scenario, showed that QCM achieves high information rates and is computationally efficient, flexible, and robust. The performance of QCM was compared to MAGIQ and other precoding algorithms including SQUID and MSM. The QCM and MAGIQ algorithms achieve the highest information rates with the lowest complexity measured by the number of multiplications. For example, Figure 4 shows that b=3 bits of phase modulation operates within 3 dB of LP-ZF. Moreover, BER simulations for a 5G NR LDPC code show that GMI is a good predictor of the coded performance. Finally, for noisy CSI the performance degradation of QCM and SQUID is qualitatively similar to the performance degradation of LP-ZF.

## Figures and Tables

**Figure 1 entropy-24-00504-f001:**
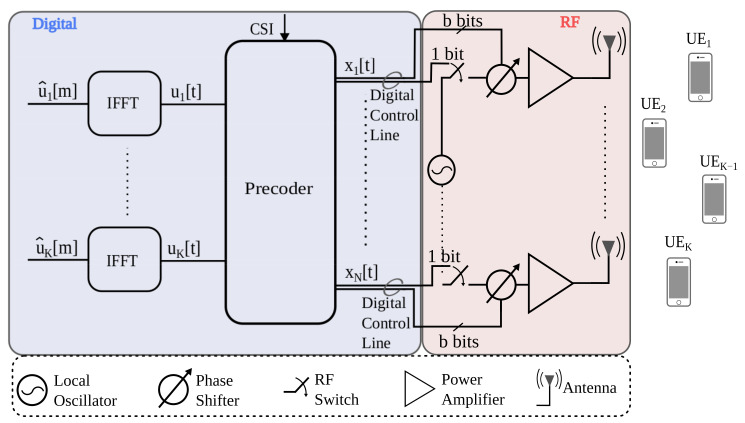
Multi-user MIMO downlink with a low resolution digitally controlled analog architecture.

**Figure 2 entropy-24-00504-f002:**
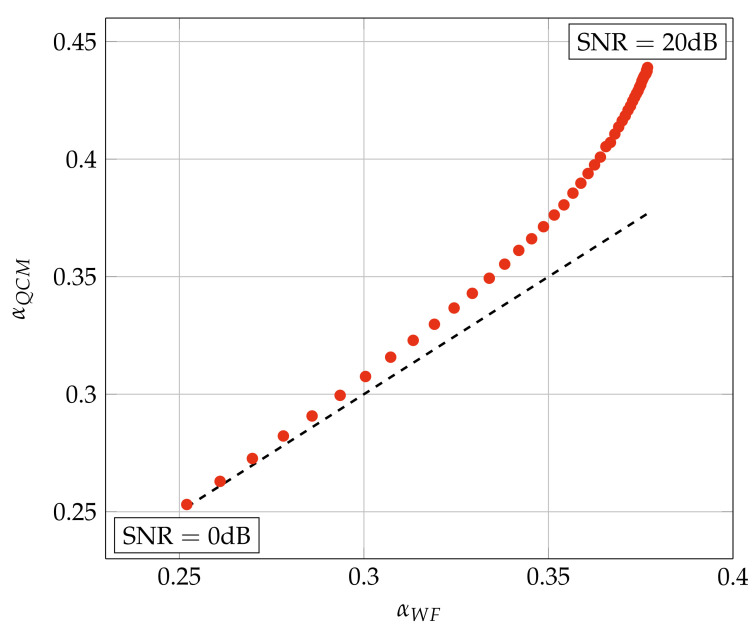
$\alpha_{\mathrm{QCM}}$ vs. $\alpha_{\mathrm{WF}}$ for System A of Table 2 and 64-QAM.

**Figure 3 entropy-24-00504-f003:**
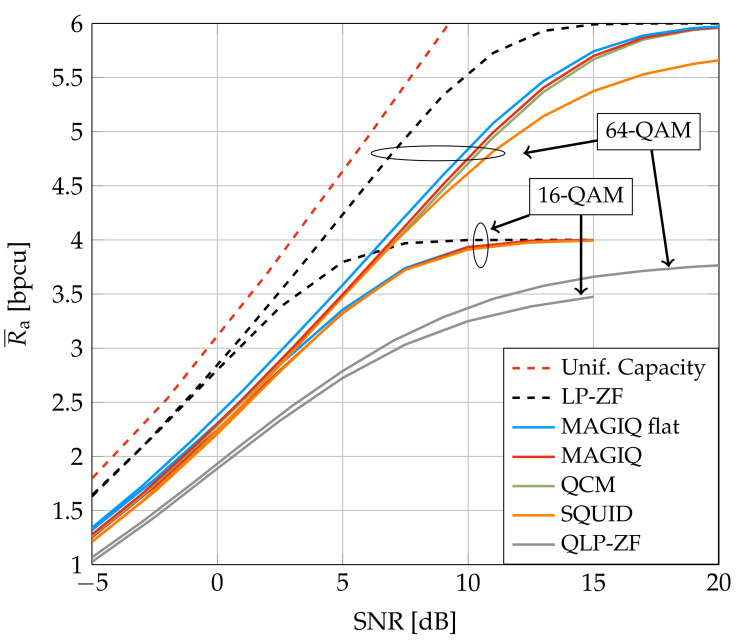
Average GMI for System A and b=2.

**Figure 4 entropy-24-00504-f004:**
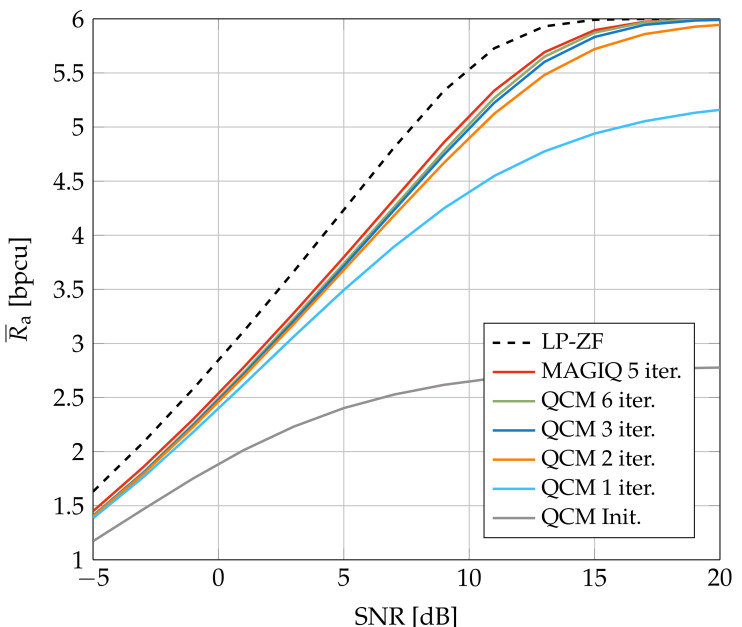
Average GMI for System A with 64-QAM and b=3.

**Figure 5 entropy-24-00504-f005:**
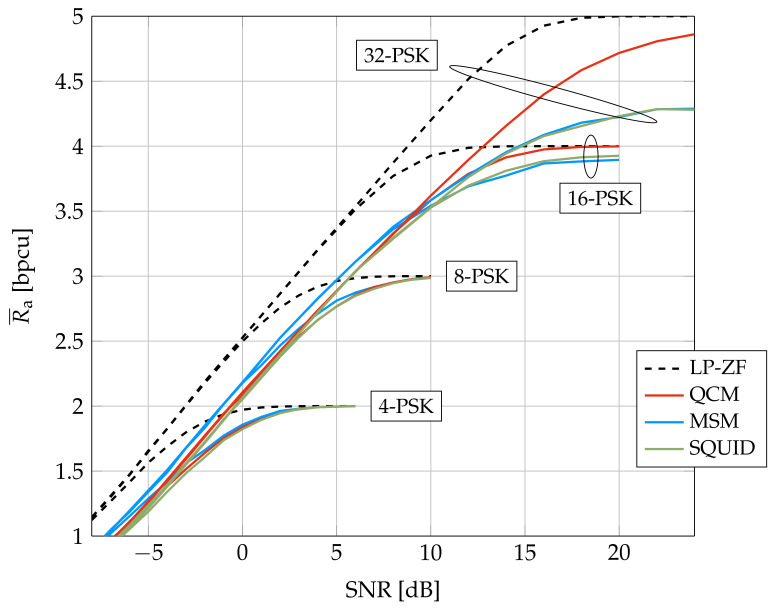
Average GMI for System B.

**Figure 6 entropy-24-00504-f006:**
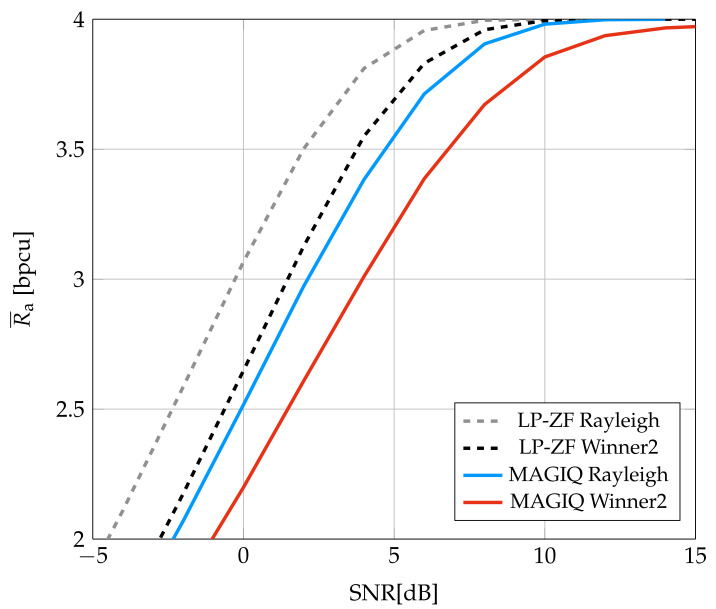
Average GMI for System C.

**Figure 7 entropy-24-00504-f007:**
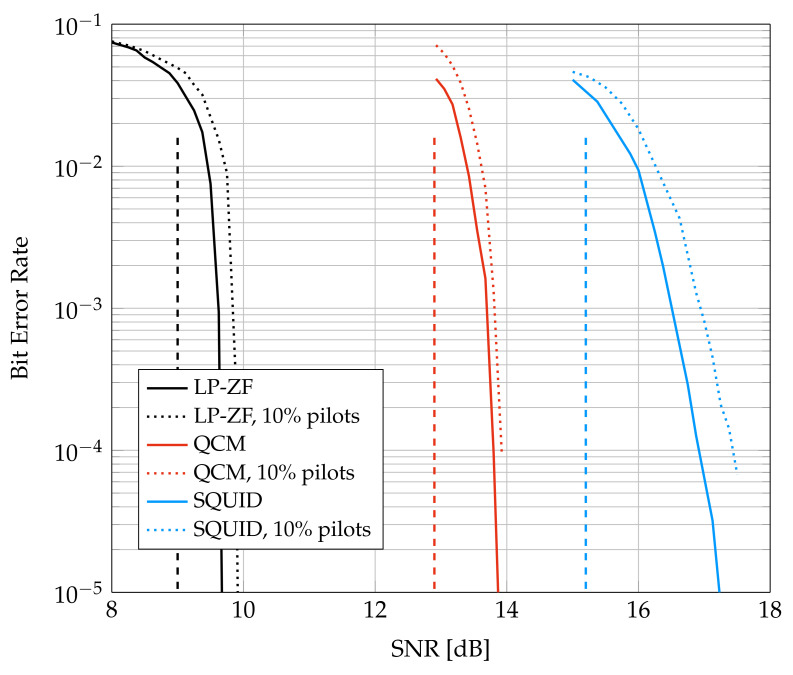
BER for System D and a 5G NR LDPC code. The dashed vertical curves show the SNR required for long random codes, see Figure 3.

**Figure 8 entropy-24-00504-f008:**
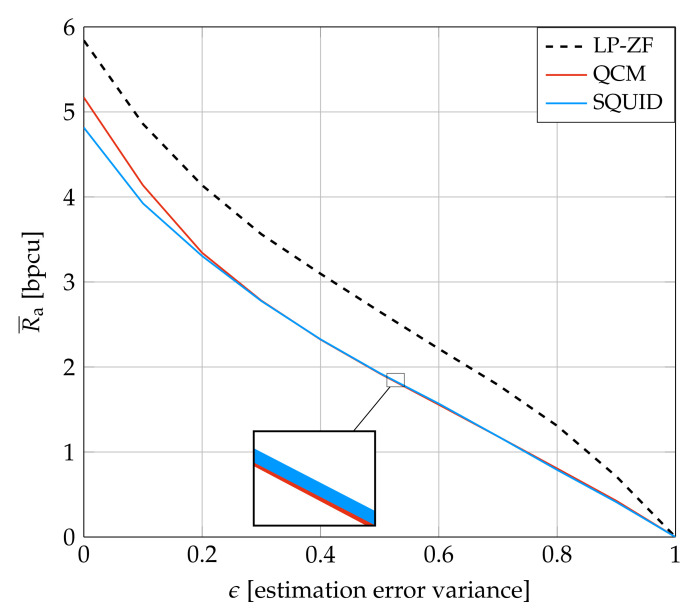
Average GMI for System A and imperfect CSI at SNR=12 dB.

**Table 1 entropy-24-00504-t001:** References for quantized precoding.

		Precoding Algorithm			
		QLP	Convex	Coord.-Wise	Other (MSM,
Modulation	Fading		Relaxation	Optimization	BB, NN, etc.)
1 Carrier	Flat	[8,9,10,11,12,13,14,15]	[16,17,18]	[19,20,21,22,23,24,25]	[26,27,29,30,31,32,33,34,35,36]
	Freq.-Sel.				[28]
OFDM	Freq.-Sel.	[37]	[38]	[39,40,41]	[42,43]

**Table 2 entropy-24-00504-t002:** System parameters for the simulations.

System	*N*	*K*	*T*	=	$T_{F}+T_{c}$	*L*	Constellation	*b*	Fading Statistics
A	128	16	270	=	256+14	15	{16, 64}-QAM	2, 3	Flat and Rayleigh
									uniform PDP
B	64	8	35	=	32+3	4	{4-32}-PSK	2	Rayleigh uniform PDP
C	80	8	277	=	256+21	22	16-QAM	2	Rayleigh uniform PDP
			286	=	256+30	varies			Winner2 NLOS C2 urban
D	128	16	410	=	396+14	15	64-QAM	2	Rayleigh uniform PDP

**Table 3 entropy-24-00504-t003:** Algorithmic complexity.

Algorithm	Multiplications per Iteration	Iterations	Pre-Processing Multiplications
QLP-ZF	$\left(TK^{3}+TK^{2}N\right)$	1	-
SQUID	(8KNT+8NTlogT)	20–300	$2T\cdot (\frac{5}{3}K^{3}+3K^{2}N+\left(6N-\frac{2}{3}\right)K)$
MSM	$\left(4KNT^{2}+4KT+2NT\right)$	≈8400	4KNT
MAGIQ & QCM	(KNTL+KNL∥)	4–6	KNT+4NTlogT

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
