# Peer review of "Low-Resolution Precoding for Multi-Antenna Downlink Channels and OFDM†"

_entropy, 2022, doi:10.3390/e24040504_

Round 1
Reviewer 1 Report
The submitted paper proposes low-resolution precoding schemes for multiuser multiple-input single-output (MU-MISO) downlink, called multi-antenna greedy iterative quantization (MAGIQ) and quantized coordinate minimization (QCM). MAGIQ was originally proposed in authors' own workshop paper [17] and QCM is a simplification of MAGIQ. The proposed schemes are compared to conventional low-resolution precoding, such as squared-infinity norm Douglas-Rachford splitting (SQUID) [16] and maximum safety margin (MSM) [10] while unquantized linear precoding with zero forcing (LP-ZF) is used as a baseline. Numerical simulations show that MAGIQ and QCM outperforms SQUID and MSM in terms of generalized mutual information and bit error ratio (BER).
The manuscript is well written and almost ready for publication. Information-theoretic analysis based on generalized mutual information is suitable for the journal Entropy. Thus, I would like to recommend acceptance for publication.
Comment
--Can the authors derive an upper bound on the mutual information (12)? A genie-aided channel estimator might work, as considered in Fig. 7. We cannot judge the superiority of the proposed schemes only by comparing lower bounds, while the superiority has been justified from the BER comparison in Fig. 7.
Author Response
We thank the reviewer for the positive comments.
Can the authors derive an upper bound on the mutual information (12)? A genie-aided channel estimator might work, as considered in Fig. 7. We cannot judge the superiority of the proposed schemes only by comparing lower bounds, while the superiority has been justified from the BER comparison in Fig. 7.
An upper bound on (12) is the capacity with uniform inputs. We added this curve to Figure 3 and provided the standard references [49,50]. Please note, however, that this does not “prove” the superiority of our QCM/MAGIQ framework, since for this we would need upper bounds on the performance of the other individual algorithms. It is not clear to us how to obtain bounds of this sort, other than the capacity. This seems to be a good topic for further research.
Reviewer 2 Report
A low-resolution QCM algorithm is proposed for multi-antenna downlink channels. The paper is well-structured.
- The reference section should be improved and updated by adding recently published papers [2018-2022]. This should be also reflected in introduction discussion.
- The authors are requested to add a comparison table to compare the fundamental characteristics of the proposed algorithm with the others reported in the literature.
- Conclusion should be rewritten.
Author Response
The reference section should be improved and updated by adding recently published papers [2018-2022]. This should be also reflected in introduction discussion. The authors are requested to add a comparison table to compare the fundamental characteristics of the proposed algorithm with the others reported in the literature.
We thank the reviewer for the suggestion. As requested, we have added about 25 more recent references. As requested, we included these in the introduction discussion, and have collected them in the comparison Table 1.
However, all the new references treat single-carrier and flat fading channels, except for [43] that only recently appeared as an early access paper. These models are substantially simpler than the multipath/OFDM channels we are considering. The few papers on OFDM were already included in our submission.
Conclusion should be rewritten.
Thank you for the remark, the conclusions have been expanded to reflect the contributions better.
Round 2
Reviewer 2 Report
The authors have addressed all the review comments. The present form of the paper can be considered for publication.